# Neuroprotection of Cannabidiol, Its Synthetic Derivatives and Combination Preparations against Microglia-Mediated Neuroinflammation in Neurological Disorders

**DOI:** 10.3390/molecules27154961

**Published:** 2022-08-04

**Authors:** Muhammad Yousaf, Dennis Chang, Yang Liu, Tianqing Liu, Xian Zhou

**Affiliations:** 1Department of Bioinformatics and Biotechnology, Government College University Faisalabad, Faisalabad 38000, Pakistan; 2NICM Health Research Institute, Western Sydney University, Westmead, NSW 2145, Australia

**Keywords:** cannabidiol, cannabidiol derivatives, neuroinflammation, microglia, neuroprotection, neurological disorders

## Abstract

The lack of effective treatment for neurological disorders has encouraged the search for novel therapeutic strategies. Remarkably, neuroinflammation provoked by the activated microglia is emerging as an important therapeutic target for neurological dysfunction in the central nervous system. In the pathological context, the hyperactivation of microglia leads to neuroinflammation through the release of neurotoxic molecules, such as reactive oxygen species, proteinases, proinflammatory cytokines and chemokines. Cannabidiol (CBD) is a major pharmacologically active phytocannabinoids derived from *Cannabis sativa* L. CBD has promising therapeutic effects based on mounting clinical and preclinical studies of neurological disorders, such as epilepsy, multiple sclerosis, ischemic brain injuries, neuropathic pain, schizophrenia and Alzheimer’s disease. A number of preclinical studies suggested that CBD exhibited potent inhibitory effects of neurotoxic molecules and inflammatory modulators, highlighting its remarkable therapeutic potential for the treatment of numerous neurological disorders. However, the molecular mechanisms of action underpinning CBD’s effects on neuroinflammation appear to be complex and are poorly understood. This review summarises the anti-neuroinflammatory activities of CBD against various neurological disorders with a particular focus on their main molecular mechanisms of action, which were related to the downregulation of NADPH oxidase-mediated ROS, TLR4-NFκB and IFN-β-JAK-STAT pathways. We also illustrate the pharmacological action of CBD’s derivatives focusing on their anti-neuroinflammatory and neuroprotective effects for neurological disorders. We included the studies that demonstrated synergistic enhanced anti-neuroinflammatory activity using CBD and other biomolecules. The studies that are summarised in the review shed light on the development of CBD, including its derivatives and combination preparations as novel therapeutic options for the prevention and/or treatment of neurological disorders where neuroinflammation plays an important role in the pathological components.

## 1. Introduction

Neurological disorders are characterised as electrical, biochemical and structural abnormalities that affect the central nervous system (CNS) and peripheral nervous system, leading to a wide-range of clinical symptoms, such as partial or complete paralysis, muscle weakness, loss of sensation, seizures and poor cognitive abilities [1]. Common neurological disorders in humans include Alzheimer’s disease (AD), Huntington’s disease, epilepsy, multiple sclerosis (MS), neuropathic pain and Parkinson’s disease [2]. It is estimated that approximately 25% of the population will develop one or more neurological disorders at some stage in life, both in developing and developed countries [1]. 

Increasing evidence showed that neuroinflammation plays a key role in the pathophysiology of common neurological disorders [3,4,5]. Neuroinflammation involves a coordinated response between microglia and other CNS cells, such as astrocytes, with the latter cells receiving and amplifying inflammatory signals from the former [6]. Microglia are the resident immune cells of the CNS, which play a crucial role in brain development and maintaining tissue homeostasis [7]. Under pathological conditions, the excessive activation of microglia in response to risk factors, such as stress, chronic inflammation or pathogens, leads to impaired neuronal survival or induced neuronal toxicity in the neighboring neuronal tissues [8]. These destructive effects in neuronal microenvironment caused by microglia-mediated neuroinflammation have been observed in many neurological disorders, such as MS, AD, hypoxia ischemia (HI), schizophrenia and neuropathic pain [8]. Thus, research to understand microglia cell-propagated neuroinflammation as a primary therapeutic target for neurological disorders has gained increasing interest [9]. 

Cannabinoids are a group of terpenophenolic compounds derived from the *Cannabis sativa* L. plant. The most potent bioactive compounds among cannabinoids include ∆8 -tetrahydrocannabinol (∆8-THC), ∆9-tetrahydrocannabinol (∆9-THC), cannabidiol (CBD) and cannabinol (CBN) [10,11,12]. CBD is a relatively well-researched bioactive compound with no psychoactive effects, unlike THC [13]. CBD, with a molecular formula of C_21_H_30_O_2_ and a molecular weight of 314.464 g/mol, is composed of twenty-one carbon atoms, thirty hydrogen atoms and two oxygen atoms (Figure 1) [14]. According to the International Union of Pure and Applied Chemistry, the name CBD refers to 2-[(1*R*, 6*R*)-3-methyl-6-prop-1-en-2-ylcyclohex2-en-1-yl]-5-pentylbenzene-1,3-diol. The naturally occurring CBD compound has a (−)-CBD [15], and the molecule of CBD comprises a pentyl side chain, a cyclohexene ring (A) and a phenolic ring (B) [16]. Recently, a drug containing a highly purified CBD, the EPIDIOLEX^®^ oral solution, has been approved by Food and Drug Administration for the treatment of seizures, which highlights the great medicinal value of CBD [17]. The therapeutic potential of CBD on neuroprotection has been investigated in numerous clinical trials for the treatment of acute and chronic neurodegenerative disorders including MS [18,19], schizophrenia [20], epilepsy [21], and autoimmune encephalomyelitis (EAE) [22], cerebral circulation during ischemic events and sepsis-related encephalitis [23,24,25]. These neuroprotective effects arise from the capability of CBD to restrain the immune responses mediated by immune cells, such as activated microglia [18] and pathogenic T-cells [26].

A number of studies have looked into the synthetic CBD derivatives for novel drug discovery with the purpose of improving efficacy, potency and pharmacokinetic properties of CBD [27]. In a pharmacological context, the synthetic CBD derivatives have been developed to bind to a variety of receptors, including peroxisome proliferator-activated receptor-γ (PPAR-γ), cannabinoid receptor 1 (CB1) and cannabinoid receptor type-2 (CB2), leading to diverse pharmacological activities, such as anti-neuroinflammatory, antitumor, anti-obesity and cardioprotective effects [12,28].

This review aims to summarise the potential anti-neuroinflammatory properties of CBD. Particular attention was given to neuroprotective effects and associated mechanisms of CBD via the inhibition of microglia-mediated neuroinflammation. Investigating the anti-neuroinflammatory potential of CBD is a significant step towards developing CBD as a novel therapeutic agent against neurological disorders. 

## 2. Effects of CBD on Neurological Disorders via Inhibiting Microglia-Mediated Neuroinflammation

A number of in vitro and in vivo studies have shown the significant neuroprotective effect of CBD in neurological disorders via targeting microglia-mediated neuroinflammation. The anti-neuroinflammatory activity of CBD was mainly manifested as reduced proinflammatory cytokines, chemokines, reactive oxygen species (ROS) and neurotoxic factors in microglia. A summary of the preclinical evidence of CBD on microglial activation and neuroinflammatory signaling pathways in various neurological disorders is listed in Table 1. 

### 2.1. Epilepsy

Epilepsy is a chronic neurological disease with recurring seizures and subsequent brain damage [40,41,42]. Neuroinflammation is commonly activated in epileptogenic brain regions in humans and animal models of epilepsy, particularly in drug-resistant epilepsy [42,43]. CBD (Epidiolex^®^, 99% CBD) is shown to attenuate seizure severity and frequency in treatment-resistant epilepsy at 5 mg/kg/day in children and adults [44,45,46], suggesting that the effect of CBD on epilepsy is at least partly related to its anti-neuroinflammatory activity. A recent animal study suggested that both prophylactic and post seizure treatment of CBD (240 mg/kg) temporarily mitigated the severity of the primary convulsive events against kainic acid-induced seizures in mice. Such protective effect from CBD was accompanied with reduced activation and the accumulation of microglia in the hippocampus and the production of iNOS post seizures. Consequently, CBD treatment was demonstrated to reduce the numbers of ectopic neurons, suggesting a suppressed neuron excitability attributed to the decreased abnormal placement of new neurons [29]. 

Although not fully understood, the anti-neuroinflammatory action of CBD may play a multi-functional role for the management of epilepsy. Cytokines, such as TNF and IL-1β, contribute to the onset of seizure activity, and increase hyperexcitability and neurotoxicity in seizure through the enhancement of Ca^2+^ influx and extracellular levels of glutamate [47,48]. Thereby, the cytokine inhibitory effect of CBD may directly contribute to its anti-seizure effect. In addition, the TNF-α suppressive effect by CBD is linked with the inhibition of adenosine reuptake of adenosine in microglia [49], which is among the determining factors for the control of seizure attacks in Dravet and Lennox-Gastaut syndromes [50]. The elevated levels of adenosine lead to the activation of presynaptic A_1_ receptors and the consequent reduced glutamate release from excitatory terminals [51]. The anti-neuroinflammatory effect of CBD is also associated with the agonist action on transient receptor potential vanilloid type 1 (TRPV1) channel. The activation of microglial TRPV1 channels is shown to mediate persistent neuroinflammation, which is at least partly responsible for the critical etiology of epilepsy [49] and the recurrent febrile seizures [52]. 

### 2.2. Multiple Sclerosis

In MS, microglia-mediated proinflammatory cytokines (i.e., IL-1β, IFN-γ and IL-12) breaks the myelin sheath in a demyelination process, and this can ultimately damage the axon, cutting the connection between neurons [53]. Three in vivo studies explored the effectiveness of CBD in various MS mouse models. Kozela et al. (2010) reported that CBD (5 mg/kg, IP) ameliorated the severity of the symptoms of EAE assessed by the histology and immunocytochemistry assays on spinal cord sections. The diminished axonal damage was accompanied by the reduced microglia activation and T-cell recruitment. The study concluded that the action of CBD was not mediated via the known CB2 receptors [37]. A latter study from Mecha M et al. (2013) suggested that the adenosine A2A receptors might be involved in the anti-inflammatory action of CBD using a vial model of MS in mice. A potent anti-neuroinflammatory activity of CBD (5 mg/kg, IP) was also shown in this study to attenuate the activation of microglia and inhibit proinflammatory cytokines (IL-1β), chemokines (CCL2 and CCL5) and the vascular cell adhesion molecule-1 (VCAM-1) [32]. Sajjadian et al. (2019) demonstrated that a CBD treatment (5 mg/kg, IP) ameliorated the cuprizone-induced demyelination in mice, and the mechanisms were associated with suppressing microglia accumulation and reducing oxidative stress [31].

### 2.3. Hypoxia Ischemia

The hypoxia-induced neuronal damage is exacerbated by the activation of microglia that produce proinflammatory cytokines, chemokines and ROS [54]. A study by Caprian et al. (2017) investigated the potential protective effect of CBD (5 mg/kg, i.p.) in a neonatal rat model with arterial ischemic stroke induced by the surgery of middle cerebral artery occlusion (MCAO) for 3 h. A set of behavioral tests was then conducted 7 days and 30 days after the MCAO-induced arterial ischemic stroke. Their results demonstrated that CBD improved neurobehavioral function in general and reduced the volume of perilesional gliosis, although it was insignificant in reducing the infarct area. Moreover, CBD reduced the microglia activation and migration, which was considered to at least partially contribute to the protective effect on neuron after HI [33].

### 2.4. Schizophrenia

Schizophrenia is a chronic brain disorder characterised by individuals interpreting reality abnormally [55]. Although the exact cause of schizophrenia remains unknown, there is a link between the hyperactivation of microglia-induced neuroinflammation and the onset of schizophrenia [56]. A study from Gomes et al. (2015) assessed the effectiveness of CBD (30 and 60 mg/kg, repeated administrations from 6th to 28th day) on glial and behavioral changes in schizophrenia in mice induced by the chronic administration of dizocilpine (MK-801). Their results demonstrated that the CBD treatments normalised the altered expressions of glial markers, including Iba-1 and GFAP in the dorsal hippocampus and behavioral disruption affected by MK-801. CBD also inhibited the activation of the microglia and astrocyte and improved neuronal function, which may contribute to the ameliorated behavioral changes in the schizophrenia mice [34].

### 2.5. Neuropathic Pain

Neuropathic pain is a neurological disorder instigated by injury or damage to the nerves that transfer signal between spinal cord and brain from the muscles, skin and other body parts [57]. Li et al. (2018) evaluated the effects of CBD on the neuroinflammatory response and recovery function in infinite horizons (HI) impactor device induced spinal cord injury in female C57B1/6 mice. In the model group, there were significant increases of proinflammatory chemokines and cytokines associated with T-cell invasion and differentiation in the injured cord. The treatment of CBD (1.5 mg/kg, IP) significantly (*p* < 0.0001) reduced the excessive neuro-inflammatory response, including IL-23, C-X-C motif ligand (CXCL)-9, CXCL-11, inducible nitric oxide synthase (iNOS) and interferon-γ (IFN-γ) in the cord. The reduced expressions of chemokine and cytokine following the CBD treatment suggested an inhibited microglial activation, although not the microglial populations. CBD attenuated neuroinflammation by suppressing the expression of proinflammatory cytokines and other neurotoxic factors, which led to a reduced neurological pain as assessed by q-PCR. Overall, the treatment of CBD attenuated the spinal cord injury, which was related to an inhibited neuroinflammation activity [35].

### 2.6. Alzheimer’s Disease

Alzheimer’s disease (AD) is a neurological disorder with the deposition of Aβ plaques as the early and invariable feature [58]. Recent studies reported that the accumulation of Aβ was closely linked with the microglial activation and the release of proinflammatory cytokines [59]. A study by Martín-Moreno et al. (2011) evaluated the effect of CBD on microglial functions in vitro and learning behaviors associated with the expressions of cytokines in Aβ accumulated mice. The results demonstrated that CBD treatment (20 mg/kg, IP) over 3 weeks significantly (*p* < 0.05) decreased the deposited Aβ peptide and improved cognitive ability in a spatial navigation task [36]. The anti-neuroinflammatory activity of CBD was associated with the reduced level of IL-6 through the modulation of microglial activation, although the effect on TNF-α gene expression was insignificant. In addition, their in vitro experiments showed a diminished nitrite level by CBD (100 nM) in cultured LPS-stimulated N13 microglial cells and rat primary microglial cells. CBD also reduced adenosine triphosphate (ATP) upsurge release and intracellular calcium (Ca^2+^), which indicated that the adenosine receptors A2A receptors may be involved in CBD action [36]. This study provides preliminary evidence to support the use of CBD as a preventative therapy for AD to alleviate AD-related symptoms, such as memory loss and behavior disorders, and the mechanism was at least partially related to the an-neuroinflammatory activity. 

## 3. Mechanisms of Action of CBD on Microglia-Mediated Neuroinflammatory Genes and Pathways

Several molecular pathways are crucial in the pathophysiology of activated microglia-mediated neuroinflammation. To begin with, the activation of toll-like receptor 4 (TLR4) of microglia stimulates the release of ROS by nicotinamide adenine dinucleotide phosphate (NADPH) oxidase, leading to the activation of NF-κB and the signal transducer and activator of transcription 3 (STAT3) signaling pathways and the subsequent production of proinflammatory cytokines, such as TNF-α, IL-6, IFN-β and IL-1β. 

A study from dos-Santos-Pereira et al. (2020) demonstrated that CBD (1–10 μM) inhibited the release of TNF-α, IL-1β, and glutamate in LPS-induced microglial cell cultures. The mechanisms were related to its antioxidant and anti-inflammatory actions. CBD exhibited an intrinsic free-radical scavenging capacity as assessed by 2,2-diphenyl-1-picryl-hydrazyl-hydrate (DPPH) assay, which in turn prevented the rise in glucose uptake against the stimulation of LPS in microglia cells. Such activity was evidence by the lowered NADPH oxidase apocynin and the downregulation of IkappaB (IκB) kinase-2. Interestingly, the downregulation of IκB kinase-2 indicated that the antioxidant capacity contributed to its anti-inflammatory activity, which has been confirmed by the subsequent downregulation of NF-κB-mediated signaling events [39]. 

In the TLR4-mediated pathways, several studies demonstrated that the NF-κB pathway plays a central role in anti-neuroinflammation of CBD. With the presence of extracellular inducers (such as LPS), the activation of upstream interleukin-1 receptor-associated kinase 1 (IRAK-1) induces the rapid phosphorylation of IκB. Consequently, both IRAK-1 and IκB undergo ubiquitination and proteosome degradation, which then activate the phosphorylation and translocation of NF-κB in the nucleus, leading to the expressions of various inflammatory genes. CBD (1, 5 or 10 μM) was found to partially reverse the degradation of IRAK-1 intermediate kinase, reverse IκB degradation, and lowered the NFκBp65 subunit phosphorylation. Thus, the reversal of the IκB- NF-kB pathway may at least partially contribute to the reduced inflammatory genes by CBD in LPS-activated BV-2 microglia [37].

The IFNβ-STAT pathway was also suggested to be involved in CBD’s anti-inflammatory action [37]. The activation of the TLR4-MyD88-dependent pathway upregulates the expression of interferon regulatory factor 3 (IRF3), which then induces the gene expressions (i.e., CXCL10, CCL5 and CCL2) via the IFN receptor and the phosphating of Janus tyrosine kinase (JAK). The phosphorylated JAK triggers the activation of STAT family, which induces the gene expressions of both pro- and anti-inflammatory genes. In particular, STAT 1 homodimers exhibit proinflammatory effects via binding to interferon-sensitive response element (ISRE) and IFN-γ-activated sequence elements. In contrast, STAT3- exerts anti-inflammatory effects via the increased synthesis of IL-10 or directly binding to consensus elements of various IL-10-inducible genes. Interestingly, it was found that CBD (1, 5 or 10 μM) dose-dependently reduced the activation of the proinflammatory STAT1 and strengthened the activation of STAT3 for the anti-inflammatory activity. This finding suggested that CBD not only decreased the proinflammatory response but also counteract inflammation, which was associated with the regulation of both STAT1 and STAT3 expressions [37].

A study from Wu et al. (2021) demonstrated that CBD (10 μM) inhibited IL-6, but not TNF-α, in LPS-stimulated mouse primary microglia and astrocytes. Interestingly, their results demonstrated that such action was not dependent on the G protein coupled receptor 3 (GPR3), a member of the G protein-coupled receptor family of transmembrane receptor, in primary mouse astrocytes and microglia [38]. 

A recent study explored the involvement of the phosphatidylinositol 3-kinase (PI3K)/protein kinase B (PKB/Akt) in the neuroprotective effect of CBD in pilocarpine microinjection-induced seizure C57Bl/6 and PI3Kγ^-/-^ mice. While CBD (30, 60 or 90 mg/kg, IP) was found to increase latency, reduce the severity of behavioral seizures and attenuate neurodegeneration and microgliosis, all the actions were abolished in the PI3Kγ^-/-^ mice. Thus, it was suggested that the PIK3 pathway played an important role in the anticonvulsant, the neuroprotective and, probably, the anti-neuroinflammatory effect of CBD [30]. 

Recent advances in research has linked the regulation of the microbiome as a major factor contributing to the attenuated neuroinflammation through the gut-microbiota-brain axis [60,61]. A study from Dopkins et al. (2021) investigated the impact of CBD on the composition of the gut microbiota in a EAE murine model. Their results demonstrated that the oral administration of CBD at 20 mg/kg resulted in no significant changes between the genera at either time point to the cecal microbiota when compared to that of the vehicle control. Furthermore, there were also insignificant changes in the short-chain fatty acids between vehicle and CBD-treated mice. Thus, although CBD significantly reduced CXCL9, CXCL10 and IL-1β expression within the CNS and inflammation in the GI tract of the EAE mice, it was considered that this was independent from the modulation of microbiome at least at that tested dosage [62].

Altogether, these studies explored the anti-inflammatory effects of CBD in microglial cells via the downregulation of TLR4-NF-κB-STAT3 pathway, the suppression of NADPH oxidase-mediated ROS expression and many other factors. A brief schematic diagram on the mechanism of action of CBD related to NF-κB and STAT-3 pathways is shown in Figure 2.

## 4. Neuroprotective Effects of Synthetic CBD Derivatives in Microglia-Mediated Neuroinflammation 

Due to promising therapeutic benefits of CBD in a variety of disorders, synthetic CBD derivatives have attracted a great attention in both academia and industry with the purpose of further improving its efficacy, potency and pharmacokinetic properties [27]. Structural modifications on CBD were generally aimed at more potent compounds with multitargeted actions than its original counterparts. The anti-neuroinflammatory potential of these synthetic CBD derivative studies are described below. The chemical structures of these synthetic CBD derivative studies are shown in Figure 3, and their anti-neuroinflammatory potential described below and shown in Table 2.

VCE-004.8 (an aminoquinone derivative of CBD) is a dual peroxisome proliferator-activated receptor-γ (PPARγ) and cannabinoid receptor type-2 (CB2) agonist produced by the air oxidation of CBD [63], with potential anti-neuroinflammatory activity [64]. A study from Navarrete et al. (2018) investigated the anti-neuroinflammatory effect of VCE-004.8 (2.5 μM) via the hypoxia-inducible factor (HIF) pathway in different types of cells. Their results showed that VCE-004.8 reduced IL-17-induced microglia polarization, inhibited LPS-induced prostaglandin E2 (PGE2) synthesis and cyclooxygenase-2 (COX-2) expression and induced the expression of arginase 1 in microglia and macrophages. Moreover, it stabilized HIF-1α and HIF-2α and activated the HIF pathway in human microvascular endothelial cells, microglial cells and oligodendrocytes. HIF-1α is involved in anti-neuroinflammatory effects by inducing the release of transforming growth factor-beta (TGF-β), which acts as a potent anti-neuroinflammatory cytokine [65]. VCE-004.8 upregulated the expression of HIF-dependent genes, such as vascular endothelial growth factor A (VEGFA) and erythropoietin, which are responsible for maintaining homeostasis and sustaining neurogenesis in adult brain [66,67]. In TMEV and EAE mouse models, a histopathological analysis showed that VCE-004.8 (10 mg/kg) treatments prevented infiltration of immune cells, as well as attenuated demyelination and axonal damage. Moreover, it downregulated the expression of several genes closely related with the pathophysiology of MS, including those contributing to the production of adhesion molecules, cytokines and chemokines [68]. These results provided a comprehensive insight about the potent role of VCE-004.8 for MS treatment by attenuating demyelination and neuroinflammation, the mechanism of which was related to the HIF pathway [69,70]. 

Dimethylheptyl-cannabidiol (DMH-CBD) is a synthetic derivative of CBD where the pentyl chain has been substituted by a dimethylheptyl chain [12]. DMH-CBD is designed to enhance the ability of a molecule to penetrate into the cell, leading to an improved bioactivity [71]. Juknat et al. (2016) assessed the anti-neuroinflammatory effects of DMH-CBD at the transcriptional level in BV-2 microglia. Their results suggested that DMH-CBD (1, 5 and 10 μM) dose-dependently downregulated the LPS-induced expressions of proinflammatory genes, including IL-1β, TNF-α and IL-6. In addition, DMH-CBD upregulated gene expressions of glutathione homeostasis, including Trb3, p8, Slc7a11/xCT, Atf4, Hmox1 and Chop, suggesting the protection of cells by inducing an adaptive cellular response against oxidative injury and inflammatory stimuli, particularly GSH deprivation [72]. Although DMH-CBD was found to target similar proinflammatory genes to those of CBD, its efficacy in comparison to that of the original CBD was not conducted. 

Abnormal cannabidiol (abn-CBD) is a synthetic derivative developed from transposition of the phenolic hydroxyl group and the pentyl side chain of CBD [12]. A recent study examined the effects of abn-CBD on the production of proinflammatory mediators and astrocyte wound-closure in LPS-induced astrocytic-microglial cocultures and astrocytic mono-culture. abn-CBD (10 μM) significantly reduced LPS-induced NO and TNF-α productions in cocultures, whereas the production of IL-6 remained unchanged. In isolated astrocytes culture, only LPS-stimulated TNF-α production was significantly decreased by the treatment of abn-CBD (10 μM). In cocultures, a significant reduction of the wound area was observed after abn-CBD treatment, whereas in isolated astrocytic cultures, lower reduction was observed without reaching a significant level. This indicated that the astrocytic wound closure was only affected by abn-CBD when microglia were present in the co-culture system [73]. This were possibly attributed to the anti-neuroinflammatory (NO and TNF-α) activity by abn-CBD in microglia but not astrocytes. The results showed that abn-CBD played a vital role in the modulation of glial neuroinflammation and astrocytic scar formation by altering the secretion of proinflammatory mediators. These findings provide a new insight for the neuroprotective effects of a promising abn-CBD for pharmacological use. 

**Table 2 molecules-27-04961-t002:** A summary of preclinical studies on the anti-neuroinflammatory activity of synthetic CBD derivatives in relation to microglial activation and neuroinflammatory signaling.

CBD Derivative	In Vitro/In Vivo Model	Key Findings	Pathways	Potential Targeted Disease/Therapeutic Potentials	Molecular Formula
VCE-004.8 [68]	MOG-induced EAE and TMEV-IDD in C57BL/6; LPS-induced BV-2 microglia and RAW264.7 cells	Ameliorated demyelination, axonal damage and neuroinflammation	Upregulated expressions of VEGFA and erythropoietin genes, reduced phenotype polarization, inhibited PGE2 and COX-2 pathways	MS	C_28_H_35_NO_3_
Dimethylheptyl- cannabidiol (DMH-CBD) [72]	LPS-induced BV-2 microglia	Decreased proinflammatory cytokines and oxidative stress-related genes	Downregulated the expressions of IL-1β, TNF-α and IL-6; upregulated the expressions Trb3, p8, Slc7a11/xCT, Atf4, Hmox1 and Chop	Not specified	C_25_H_38_O_2_
Abn-CBD [73]	LPS-activated astrocytic-microglial cocultures and astrocytic mono-culture	Reduced glial neuroinflammation and promoted astrocytic scar formation	Reduced productions of TNF-α and NO	Not specified	C_21_H_30_O_2_

## 5. Enhanced Anti-Neuroinflammatory Effects of CBD in Combination with Other Compounds

A few preclinical studies demonstrated that CBD combined/conjugated with some other key cannabinoids within cannabis or an antimalaria drug may exhibit an enhanced anti-neuroinflammatory activity and reduced toxicity (Table 3). The knowledge may encourage a further evaluation of the potential synergistic mechanism of key bioactive compounds within cannabis and therapeutic drugs and promote the pharmaceutical development of new and optimized compound-formula from cannabis for the treatment of neurological disorders via the targeting of neuroinflammation.

### 5.1. CBD in Combination with THC against MS

Two studies by Al-Ghezi et al. (2019) demonstrated that the combination of THC and CBD treatment (10 mg/kg each) significantly attenuated EAE related clinical signs in MS. In the first study, treatment with the combination of THC and CBD caused a significant reduction in brain infiltrating CD4+ T cells and proinflammatory molecules, including IL-6, IL-17A, TNF-α, IL-1β and IFN-γ, while enhancing anti-inflammatory phenotypes, such as IL-4, IL-10, TGF-β, STAT5b and FoxP3. The miRNA micro-array analysis using the CD4+ T cells derived from the brain showed that the combination of THC and CBD significantly downregulated the expressions of miRNAs, including miR155-5p, miR-146a-5p, miR-21a-5p, miR-122-5p, miR-31-5p, miR-27b-5p and miR-150-5p, while upregulated miR-7116 and miR-706-5p. The modulation of a series microRNAs may contribute to the observed anti-neuroinflammatory activity of THC and CBD combination. It was worth mentioning that only the combination of THC and CBD suppressed neuroinflammation and attenuated EAE development, whereas the individual treatment of CBD (10 mg/kg) or THC (10 mg/kg) alone did not cause any significant suppression of clinical symptoms of EAE [74]. In another study, the combination of THC and CBD (10 mg/kg each) significantly reduced proinflammatory cytokines, such as IFN- γ and IL-17, as well as promoted the induction of anti-neuroinflammatory cytokines TGF- β and IL-10 in EAE mice. The 16S rRNA sequencing of bacterial DNA extracted from the gut showed that EAE mice displayed higher abundance of mucin degrading bacterial species, such as *Akkermansia muciniphila*, compared to that of the blank control group, which was significantly reversed following the THC + CBD treatment. The restored abundance and diversity of gut metabolome were considered contributing to the restored levels of short chain fatty acids, such as isovaleric, valeric and butyric acids, which may result in the reduced production of proinflammatory cytokines and attenuated EAE symptoms [75]. Collectively, the results from these studies suggested that combined THC and CBD treatment may provide a potential superior therapeutic approach to using CBD or THC alone for the treatment of MS individuals in the future by downregulating miRNAs and altering the gut microbiome. 

### 5.2. CBD in Combination with Beta-Caryophyllene against Ischemia

Beta-caryophyllene (BCP) is a bioactive compound found in Cannabis with potent and diverse pharmacological properties, including antimicrobial, cardioprotective, antioxidant, neuroprotective and anti-neuroinflammatory effects [76]. A recent study by Yoqubaitis et al. (2021) examined the combined effect of CBD and BCP on permanent ischemia without reperfusion in a mouse model of photothrombosis. The mice were treated with CBD (3–30 mg/ kg, IP) and BCP (3–30 mg/kg, IP) alone or in combination (3:30, 10:30 and 30:30 CBD:BCP, *w/w*). The results showed that the CBD and BCP combinations significantly (*p* < 0.05) suppressed the microglia activation manifested by the Iba-1 immunofluorescence as compared to the model group and each monotherapy group, whereas the effect of CBD or BCP alone was insignificant. In addition, the individual BCP or CBD treatment significantly (*p* = 0.005) reduced infarct size, while a greater therapeutic efficacy was seen in the combined treatment, which showed a narrow and shallow infarct size. In addition, the combination significantly (*p* < 0.05) improved motor performance in comparison to the model group in the behavior test. Noticeably, the combination showed more potent overall therapeutic effects against ischemia compared to each single administration, suggesting that these phytocannabinoids in combination showed an improved clinical outcome against ischemia via further suppressed microglia activation [77]. 

### 5.3. Enhanced Anti-Neuroinflammatory Effects of CBD in Combination with Dihydroartemisinin

Dihydroartemisinin (DHA) is an antimalarial drug synthetical derived from natural artemisinin, which has been shown to possess potential pharmacological properties, such as anticancer, antimalarial and anti-neuroinflammatory [78]. A study from Wang et al. (2021) investigated the effectiveness of CBD and DHA alone and in mixture, or CBD-DHA conjugates (C2D, C3D, C4D) in LPS-stimulated BV-2 microglia cells. The interaction of combining CBD and DHA as two individual unlinked drugs were additive, as assessed by the combination index model (combination index = 1) in inhibiting LPS-stimulated NO production. Moreover, the combination of CBD and DHA damaged the BV-2 cellular viability, suggesting that the combination induced significant cytotoxicity. In contrast, the conjugate C3D showed a minimal cytotoxicity without significantly compromising the anti-neuroinflammation activity. The C3D also showed a greater potency in inhibiting the -stimulated NO, IL-1β and mRNAs of iNOS, as compared to those of the C2D and C4D conjugates. The signalling characterization showed that C3D inhibited NF-κB but not MAPKs activation, therefore blocking neuroinflammation in LPS-induced BV-2 microglia [79]. The results from this study supported the combination by conjugation as a promising approach to improve the therapeutic index of CBD, as evidenced by the increased therapeutic index in comparison to using the conventional combination therapy alone. 

**Table 3 molecules-27-04961-t003:** Anti-neuroinflammatory effects of CBD/CBD conjugates with other bioactive compounds in Cannabis and pharmaceutical drugs.

Combination Preparations of CBD	In Vitro/In Vivo Model	Key Findings	Modulated Neuroinflammatory Mediators	Potential Targeted Disease/Therapeutic Potentials
CBD-THC [74]	MOG-induced EAE in C57BL/6 female mice	Decreased neuroinflammation in murine EAE	Modulated the expression of miRNAs-mediated signalling pathways. Reduced CD4+ T cells, IL-6, IL-17, IL-1β, TBX21 and INF- γ, increased IL-4, IL-10, TGF-β, STAT5b and FoxP3	MS
CBD-THC [75]	MOG-induced EAE in C57BL/6 female mice	Attenuated EAE severity and reduced neuroinflammation	Decreased the levels of IFN-γ and IL-17, increased the levels of TGF-β and IL-10, modulated gut microbiota	MS
CBD-BCP [77]	Photothrombosis-induced permanent ischemia in C57B/6 male mice	Improved motor performance, reduced infarct size, modulated microglial activation	Not determined	Permanent ischemia
CBD-DHA conjugation [79]	LPS-stimulated BV-2 microglia	Blocked neuroinflammation, eliminated neurotoxicity, improved therapeutic index	Inhibited NF-kB, IL-1β, iNOS and NO	Not specified

## 6. Discussions

CBD is one of the most potent bioactive compounds among cannabinoids, which has been shown clinical benefits against neurological disorders in a number of clinical trials [47,80,81,82,83,84,85,86,87,88,89,90,91,92,93,94,95]. However, most of their clinical outcomes did not provide the link of the efficacy of CBD to the parameters related to microglia-mediated neuroinflammation, which is deemed as an important therapeutic target of the disorders. This present review focuses on the pharmacological action of CBD targeting microglia-mediated neuroinflammation, which may partially explain the mechanism of CBD in attenuating the clinical symptoms of neurological disorders. 

Three in vitro studies showed the anti-neuroinflammatory effects of CBD in LPS-induced microglia. Key molecular pathways, such as NF-κB and STAT3, were investigated to illustrate how CBD acts in order to attenuate the microglia-mediated neuroinflammation in these studies at the molecular level. Moreover, a number of key molecular targets of CBD, such as ROS and NADPH oxidase, were recognised, which are probably directly relevant to the anti-neuroinflammatory effects of this compound in preclinical and clinical studies. However, most of the in vitro studies are largely carried on single cell line, which does not take into account of the cross talk of neuronal cells within the CNS microenvironment. Although key molecular targets of neuroinflammation were evaluated, the evidence on neuroprotection against nerve damage and the prevention of the degeneration of the CNS via anti-neuroinflammatory activity remains unexplored [96,97]. In this regard, the drugs, which showed a potent efficacy in single cell line does not necessarily lead to a comparable therapeutic effect in animal models and humans. 

Nine in vivo studies supported the anti-neuroinflammatory effects of CBD in different disease models, such as epilepsy, MS, Schizophrenia, neuropathic pain, HI and AD. These studies also demonstrated the anti-neuroinflammatory activity via targeting the key proinflammatory cytokines, including TNF-α, IL-1β, IL-6 and NO. However, these studies did not show a clear link between neuroinflammation and neuroprotection. Thus, further research may advance the knowledge in investigating the clear link between the anti-neuroinflammation by CBD and the observed improvement of the relevant clinical symptoms. 

Three preclinical (two in vitro and one in vivo) studies investigated the anti-neuroinflammatory effects of synthetic CBD derivatives. These studies demonstrated that synthetic CBD derivatives exhibited potent therapeutic effects in suppressing neuroinflammation. Although a number of preclinical studies have shown interesting pharmacological properties of these derivatives, few CBD-derivatives has yet been introduced into clinical trials. This is perhaps due to the lack of evidence that the efficacy of synthetic CBD derivatives is superior to that of the original CBD. Thus, further research remains necessary to develop CBD derivates that can show greater therapeutic value than that of CBD. The constant development of synthetic derivatives continues to hold considerable potential for the discovery of novel pharmaceuticals. 

Three in vivo studies investigated the interaction of CBD with other compounds in cannabis, which both demonstrated the enhanced bioactivity in inhibiting microglia-mediated neuroinflammation. These findings suggested a promising synergistic interaction among bioactive compounds within cannabis, which may enhance the future research in isolating active fractions from cannabis that possess greater efficacy than using a full-spectrum of cannabis or a single bioactive compound. In addition, one in vitro study investigated the drug–drug conjugate approach to enhance anti-neuroinflammatory effects of CBD, which showed a minimal cytotoxicity without significantly compromising the anti-neuroinflammation activity in comparison to simply combing two compounds as unlinked drugs. This finding may encourage the novel approach for the preparation of CBD active fractions using drug conjugation technology. The findings from these studies may present a great commercial significance, as many pharmaceutical companies are actively searching for the best version of cannabis that only contain the therapeutic compounds and remove those unfavored ones that may bring side effects. Studies that showed synergistic anti-neuroinflammatory activities when combing CBD with THC or BCP highlighted the promising synergy within the cannabis, which may offer the novel pharmaceutical development from isolating active fractions from cannabis. 

Whilst mounting evidence supports the therapeutic use of CBD against neurological disorders, the safety aspects should not be neglected. For instance, CBD has shown therapeutic efficacy for Lennox-Gastaut and Dravet syndromes and is possible to have it prescribed off-label by clinicians for other conditions [98,99]. However, a few preclinical and clinical studies reported toxicity and adverse effects following CBD administration. In animals, a few adverse effects of CBD administration have been reported, including hepatocellular injuries, developmental toxicity, CNS inhibition, spermatogenesis reduction, hypotension and male reproductive alteration [100]. In humans, CBD studies for epilepsy and other neurological disorders reported CBD-induced drug–drug interactions, diarrhea, vomiting, fatigue, somnolence and hepatic abnormalities [101]. Moreover, potential drug–drug interactions and adverse effects regarding CBD-use with other compounds/drugs must be taken into consideration by physicians before recommending off-label CBD [102]. Thus, more toxicity and clinical studies are required to evaluate the possible toxicity and adverse reactions of CBD while treating neurological disorders. 

## 7. Conclusions

The preclinical studies summarised in this review supported the therapeutic use of CBD in treating neurological disorders from its action in addressing microglia-mediated neuroinflammation. The findings of this review shed light on the development of CBD and relevant compounds as novel and more advantageous therapeutics to prevent or treat neurological disorders by targeting microglia-mediated neuroinflammation. 

## Figures and Tables

**Figure 1 molecules-27-04961-f001:**
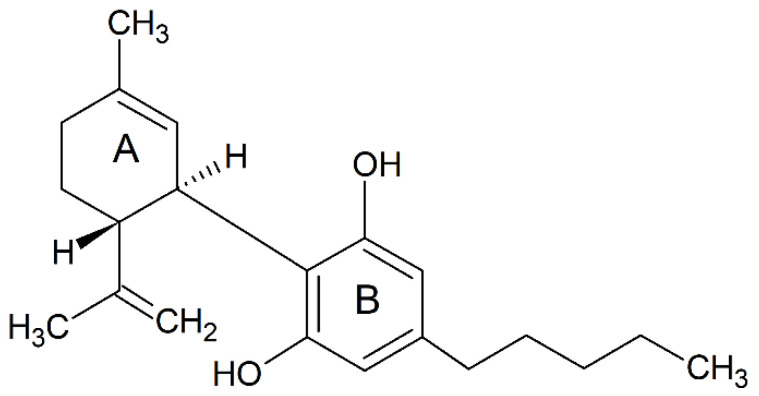
The chemical structure of the naturally occurring CBD (drawn from ACD/ChemSketch), [14]. A: cyclohexene ring. B: phenolic ring.

**Figure 2 molecules-27-04961-f002:**
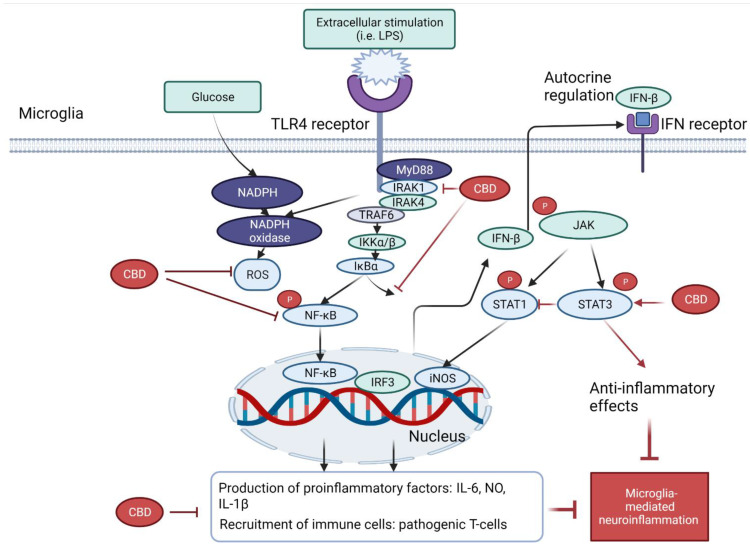
Schematic diagram of anti-neuroinflammatory effects of CBD related to NF-κB and STAT-3 pathways. The activation of TLR4 of microglia stimulates the production of ROS by NADPH oxidase, leading to the activation of the NF-kB and STAT1 signalling pathways and the subsequent production of proinflammatory mediators, such as TNF-α, IL-6, IFNβ and IL-1β. On the other hand, MyD88 activation also stimulates the uptake of glucose and synthesis of NADPH, which led to oxidative stress (ROS). CBD was found to minimize the oxidative stress and reduced the release of proinflammatory mediators through the inhibition of NADPH oxidase and modulating NF-kB/STAT signalling pathways.

**Figure 3 molecules-27-04961-f003:**
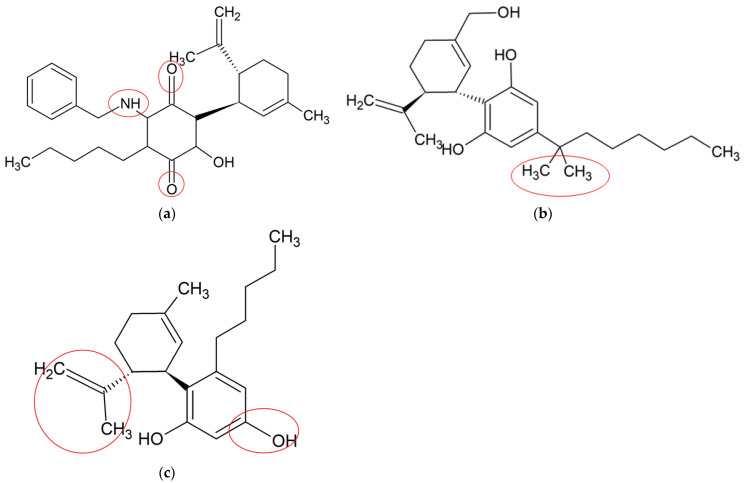
Chemical structures of three synthetic CBD derivatives, (**a**) VCE-004.8, (**b**) DMH-CBD and (**c**) Abn-CBD, which have shown anti-neuroinflammatory potentials. The structural modifications are highlighted by red circles.

**Table 1 molecules-27-04961-t001:** A summary of preclinical evidence of CBD on microglial activation and neuroinflammatory signaling.

Potential Targeted Disease/Therapeutic Potentials	Subjects	Key Findings on Clinical Biomarkers/Signs	Mechanism of Actions in Relation to Anti-Neuroinflammatory
Epilepsy	Kainic acid-induced mice with seizures [29]	Temporarily attenuated seizure scores	Inhibited inflammatory microglia activation and accumulation, reduced inducible nitric oxide synthase (iNOS) expression and decreased numbers of ectopic neurons
Epilepsy and neuroprotection	Bilateral intra-hippocampal pilocarpine microinjection-induced seizures in C57Bl/6 wild-type (WT) and PI3Kγ^-/-^ mice; Glutamate-induced primary neurons [30]	Increased latency and reduced the severity of behavioral seizures, prevented neurodegeneration, microgliosis and astrocytosis in wild type animals but not in PI3Kγ^-/-^; PI3Kγ inhibition or deficiency abolished CBD’s neuroprotective effect on glutamate-induced cell death	Involvement of phosphoinositide 3-kinase (PI3K)/protein kinase B (Akt)/mechanistic target of rapamycin (mTOR) signaling pathway
MS	Cuprizone-induced MS in male C57BL/6 mice [31]	Ameliorated demyelination	Reduced microglia accumulation and oxidative stress (catalase, superoxide dismutases and glutathione)
MS	Theiler’s murine encephalomyelitis virus-induced demyelinating disease (TMEV-IDD)-susceptible female SJL/J mice model [32]	Attenuated motor deficiencies	Reduced proinflammatory cytokines and chemokines, inhibited microglial activation and downregulated expressions of vascular cell adhesion molecule-1, chemokine (C-C motif) ligand (CCL)2,CCL5, tumor necrosis factor (TNF)-α and interleukin (IL)-1β
MS	Myelin oligodendrocyte glycoprotein (MOG)-induced EAE in C57BL/6 mice model, MOG induced encephalitogenic T-cell line [26]	Ameliorated clinical signs of EAE, reduced axonal damage and neuroinflammation, inhibited microglial activation	Inhibited T-cell proliferation, downregulatedexpression of ionized calcium binding adaptor molecule 1 (Iba-1) and galectin-3 (Mac-2) in the spinal cord
HI	Middle cerebral artery occlusion (MCAO)-induced HI in Wistar rats [33]	Improved functional recovery and reduced brain damage	Reduced apoptosis and neuronal loss; inhibited neuroinflammation, reduced microglial and astrogliosis activation and proliferation
Schizophrenia	Dizocilpine (MK-801) induced schizophrenia in male C57BL/6 mice [34]	Ameliorated behavioral changes	Modulated expressions of Iba-1 (a microglia marker), glial fibrillary acidic protein (GFAP, an astrocyte marker) and neuronal nuclear antigen (a neuronal marker)
Neuropathic pain	HI induced-spinal cord injury in female C57Bl/6 mice [35]	Attenuated neuropathic pain and high thermal sensitivity	Reduced proinflammatory cytokines and chemokines, including IL-23, CXCL-9, CXCL-11, iNOS and interferon gamma
AD	β-amyloid-injected AD C57/BI6 mice; lipopolysaccharides (LPS)-induced BV-2 and N13 microglia [36]	Improved learning in a spatial navigation task, decreased deposited Aβ peptide	Suppressed neuroinflammation in AD animal model, reduced intracellular calcium (Ca^2+^), inhibited nitric oxide (NO)-modulated microglial activation, decreased IL-6 gene expression
In vitro investigations	LPS-induced BV-2 microglia [37]	Reduced neuroinflammatory cytokines, such as IL-1β, IL-6 and interferon-beta (IFN-β)	Downregulated nuclear factor kappa B (NF-kB) signaling pathway
	LPS-induced microglia and astrocytes (host not specified) [38]	Ameliorated activation of microglia and astrocytes, inhibited production of IL-6 from microglia and astrocyte cell line	Inhibited phosphorylation of NF-κB and signal transducer and activator of transcription 3 (STAT3) signalling pathways
	LPS-induced microglia in mice [39]	Reduced synthesis of glucose-derived nicotinamide adenine dinucleotide phosphate hydrogen (NADPH) and reduced neuroinflammatory mediators (ROS and TNF-α)	Downregulated NF-kB-dependent signaling pathway

## Data Availability

Not applicable.

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
