# Peer review of "Neuroprotection of Cannabidiol, Its Synthetic Derivatives and Combination Preparations against Microglia-Mediated Neuroinflammation in Neurological Disorders"

_molecules, 2022, doi:10.3390/molecules27154961_

Round 1

Reviewer 1 Report

    I have read the manuscript entitled “Neuroprotection of cannabidiol, its synthetic derivatives and combination preparations against microglia-mediated neuroinflammation in neurological disorders” and I think that it is interesting review that describes some possible mechanisms of protective action of cannabidiol in some neurological diseases. The Authors focus on the mechanisms related to microglia-mediated neuroinflammation.
    Although in the manuscript the Authors indicated that cannabidiol is clinically used in the treatment of numerous neurological disorders, namely Alzheimer’s disease, Hungtington’s disease, epilepsy, multiple sclerosis, neuropathic pain, and Parkinson’s disease, only some of these diseases are included in the detailed descriptions of the mechanism of action of cannabidiol. The effect of cannabidiol on microglia-mediated neuroinflammation processes associated with epilepsy, Parkinson's disease, or Huntington's disease was not considered in the manuscript. In my opinion, it is essential to discuss these mechanisms of action of cannabidiol in epilepsy, as it is one of the diseases for which cannabidiol is increasingly used in therapy.
I think that the manuscript is a valuable source of knowledge on the effect of cannabidiol on microglia-mediated neuroinflammatory processes associated with various neurological diseases, and supplementing it with additional information will improve its quality and usefulness. After completion, I think the manuscript could be published in Molecules and will be of interest to the readers of this journal.

Author Response

Dear Reviewer, 

Thank you so much for your kind comments. We initially did not include the studies on epilepsy, Parkinson's disease, or Huntington's disease because there was no clear link between CBD's treatment of these diseases and its anti-neuroinflammatory action. 

As suggested, we have conducted a thorough search on CBD in treating epilepsy through anti-neuroinflammation. In the revised manuscript, we have included a section on CBD in treating epilepsy via its anti-neuroinflammatory activity (see section 2.1 Epilepsy). We have also elaborated the section on mechanisms of action to include the associated mechanism of CBD for epilepsy (lines 267-273). 

Many thanks again for your consideration of this manuscript. 

Kind regards,

Xian on behalf of all authors.  

Reviewer 2 Report

In this manuscript the authors were summarised of Neuroprotection of cannabidiol and its synthetic derivatives used in the synthesis of bioactive compounds handle to explore medicinal applications and Cannabidiol is one of the two best-known active compounds, it was phytocannabinoid with therapeutic properties for numerous disorders the potential anti-neuroinflammatory properties of CBD. The neuroprotective potential of CBD, based on the combination of its anti-inflammatory and anti-oxidant properties, intensely preclinical research in numerous neurodegenerative disorders Investigating the anti-neuroinflammatory potential of CBD is a significant step for developing CBD as a novel therapeutic agent against neurological disorders.  CBD against various neurological disorders with a particular focus on their main molecular mechanisms of action which were related to the down regulation of NADPH oxidase mediated ROS, TLR4-NFκB and IFN-β-JAK-STAT pathways, these results can be summarized in the following  previous reviews based on the development of CBD as a potential novel therapeutic option for the prevention and/or treatment of neurological disorders where neuroinflammation plays an important role in the pathological components.

I suggest accepting this manuscript for publishing in this journal

Author Response

Dear Reviewer, 

Thank you so much for the endorsement of our manuscript. 

All the best,

Xian on behalf of all authors.